# Is negative capacitance FET a steep-slope logic switch?

Wei Cao [1] & Kaustav Banerjee [1]*

The negative-capacitance field-effect transistor(NC-FET) has attracted tremendous research efforts. However, the lack of a clear physical picture and design rule for this device has led to numerous invalid fabrications. In this work, we address this issue based on an unexpectedly concise and insightful analytical formulation of the minimum hysteresis-free subthreshold swing (*SS*), together with several important conclusions. Firstly, well-designed MOSFETs that have low trap density, low doping in the channel, and excellent electrostatic integrity, receive very limited benefit from NC in terms of achieving subthermionic *SS*. Secondly, quantum-capacitance is the limiting factor for NC-FETs to achieve hysteresis-free subthermionic *SS*, and FETs that can operate in the quantum-capacitance limit are desired platforms for NC-FET construction. Finally, a practical role of NC in FETs is to save the subthreshold and overdrive voltage losses. Our analysis and findings are intended to steer the NC-FET research in the right direction.

---

[1] Department of Electrical and Computer Engineering, University of California, Santa Barbara, CA 93106, USA. *email: kaustav@ece.ucsb.edu

  **1**

Although the global semiconductor industry has reached a staggering ~$500 billion, to further continue such a flourishing growth, low-voltage or steep-slope logic transistor is crucial for simultaneous reduction of power consumption along with dimensional scalability of the metal oxide semiconductor field-effect transistors (MOSFETs)[1,2]. However, lowering of the supply voltage, has failed to keep pace with the feature-size scaling because the thermionic emission based transport mechanism of MOSFETs limits the steepness of their transfer characteristics in the subthreshold regime, i.e., SS, to be ≥ 60 mV per decade at room temperature[3], thereby constraining the energy/power efficiency. To overcome this limitation, many steep-slope (SS < 60 mV per decade) device concepts have been proposed, such as tunnel FETs[4,5], impact ionization MOSFETs[6], NC-FETs[7] etc., (Supplementary Note 1). Among these devices, NC-FETs have been attracting most interest in recent years[7–28], thanks to their simple structure that involves minimal penalty in added (w.r.t. MOSFETs) manufacturing complexity. The NC layer can be implemented with single-domain (best case) ferroelectric (FE) materials, which are featured by their "double-well" energy landscape versus polarization[29] (see Supplementary Note 2 and Supplementary Fig. 2a). The anomalous polarization response of FE material systems in the metastable state to external electric field, results in an effective NC (see Supplementary Fig. 2b). It has been suggested[7] that connecting an FE layer in series with a normal dielectric (DE) capacitor with a proper capacitance matching between FE and DE can stabilize (see Supplementary Fig. 2c) the metastable NC state of the FE layer, which has been experimentally verified recently[19,20]. It was also predicted that NC-FETs may be plagued by hysteresis, if the capacitance matching process is inappropriately performed, which should be avoided in logic applications. Driven by the enthusiasm toward this novel "steep-slope" device, experimentalists have been indiscriminately trying to incorporate FE materials in the gate of various "X-FETs", where X can be SOI, Fin, nanowire (NW), 2D (such as MoS₂, WSe₂ etc.), or 1D (such as carbon nanotube (CNT), graphene nanoribbon (GNR) etc.), generating numerous "NC-FETs" in the past several years[8–15]. However, none of them exhibit SSs that are noticeably smaller than 60 mV per decade and simultaneously free of hysteresis during both AC and DC operations (see Supplementary Table 1). Such a deviation from expectation stems from a lack of clear understanding of NC-FETs for many device experimentalists. In this sense, comprehending what NC-FETs really are, how to design NC-FETs, and what can be expected from NC-FETs, can play a decisive role in identifying a genuinely low-voltage switch that can revolutionize next-generation electronics including computing, sensing, and communication.

Previous attempts at understanding the workings of NC-FETs either assumed the total FET gate capacitance to be a constant[7,18], i.e., the bias-dependent channel capacitance was ignored, or were focused on performing intensive numerical simulations[21–25] or complex analytical modeling[26,27] of a specific NC-FET such as NC-FinFET and NC-2DFET etc., in which the generic device physics, design rule and major challenges of NC-FETs get obscured. In this work, we analytically formulate the slope of the transfer characteristics of a generic NC-FET structure, from which a physics-rich but concise picture is innovatively extracted for "*visualizing*" the intriguing device physics and design space of NC-FETs. This picture can not only serve as a convenient design platform for NC-FET experimentalists, but also allow us to find a novel functionality of NC in modern MOSFET applications.

More specifically, using fundamental FET and NC theory, we uncover the intriguing interplay between hysteresis and subthermionic (or < 60 mV per decade) SS in NC-FETs in a very simple and generic manner, and arrive at several important conclusions that are contrary to many contemporary understandings on NC-FETs. Our analysis reveals that well-designed scaled MOSFETs that have low trap density and low doping in the channel, and excellent electrostatic integrity by employing state-of-the-art FET structures, such as SOI-FET, FinFET, NW-FET, CNT-FET, and 2D material based FETs etc., which have negligible parasitic capacitance compared to the gate capacitance, receive very limited benefit from NC in terms of achieving small SS (< 60 mV per decade). Secondly, we derive an unexpectedly concise and insightful formula for the minimum SS for hysteresis-free operation, which allows us to uncover that the quantum capacitance is the limiting factor for NC-FETs to achieve hysteresis-free subthermionic SS, and thus, only FETs that can operate in the quantum-capacitance limit are desired platforms for NC-FET construction. Thirdly, our analysis unveils a hitherto concealed knowledge that compared to the target of achieving subthermionic SS, a more practical role of NC in FETs could be to save the subthreshold voltage loss caused by short-channel effects, and the overdrive voltage loss caused by the large quantum capacitance, provided the polarization response of ferroelectric materials can be enhanced to allow the NC to function effectively at high frequencies (GHz range).

## Results

**Capacitor network in a generic NC-FET.** The switching of FETs is realized by electrostatic (or capacitive) modulation of the potential of the channel through which current is conducted. Figure 1a depicts all relevant capacitors in a generic NC-FET structure. The discussion throughout this work is based on n-type

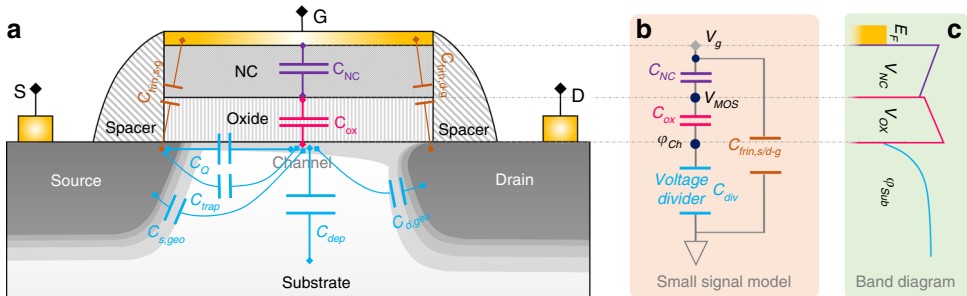

**Fig. 1 Capacitors in a generic NC-FET structure and its small-signal model. a** Capacitors potentially involved in determining the channel potential of a generic NC-FET. $C_Q$, $C_{dep}$, $C_{s/d,geo}$, and $C_{trap}$ are the quantum-, depletion-, s/d geometrical, and trap induced, capacitors, respectively. $C_{NC}$ and $C_{OX}$ represent the capacitance of the NC and oxide layers, respectively. **b** Simplified small-signal model for NC-FETs. Note that in the small-signal model, the change in the values of the capacitances near the vicinity of a given DC bias point is negligible. $C_{div}$ is the voltage divider capacitance that includes $C_Q$, $C_{dep}$, $C_{s/d,geo}$, and $C_{trap}$. The effect of fringing capacitance, $C_{frin,s/d-g}$, is screened by power rails, and hence is irrelevant to channel potential control. **c** The conduction band diagram of NC-FETs under a positive gate bias. Note that due to negative $C_{NC}$, $V_{NC}$ is negative.

device, if not specified otherwise. $C_{NC}$ and $C_{OX}$ are the capacitances of the NC layer and gate oxide, respectively. $C_Q$, $C_{trap}$, $C_{dep}$, and $C_{s/d,geo}$ represent quantum, interface trap induced, depletion, and source/drain geometrical capacitances, respectively, and can be grouped as gate voltage divider capacitance $C_{div}$,

$$C_{div} = C_Q + C_{trap} + C_{dep} + C_{s,geo} + C_{d,geo} \quad (1)$$

$C_Q$ in FETs describes the change of electron charge density ($Q_e$) in response to channel potential ($\varphi_{Ch}$) modulation, and can be expressed as

$$C_Q = \frac{dQ_e}{d\varphi_{Ch}} = q^2 \cdot \sum_i \frac{DOS_i}{1 + exp\left(\frac{E_{c,i} - E_f}{kT}\right)} \quad (2)$$

where $q$ is the elementary charge, $i$ is the index of conduction modes, $DOS_i$ is the density-of-states of the $i^{th}$ mode, $E_{c,i}$ is the energy level of the $i^{th}$ mode, and $E_f$ is the Fermi level. $C_Q$ approaches zero below threshold, while approaches $q^2 \cdot \Sigma_i DOS_i$ above threshold. $C_{trap}$ is the trap state filling induced capacitance (~$dQ_{trap}/d\varphi_{Ch}$). $C_{dep}$ is the geometrical capacitance of the depletion layer. $C_{s/d,geo}$ stem from the capacitive coupling from source and drain. This capacitance also captures the effect of fringing field from source/drain toward gate through the path inside the device in the short-channel condition[28]. As indicated in the small-signal diagram in Fig. 1b, $C_{div}$ prevents the channel potential $\varphi_{Ch}$ from being efficiently modulated by the gate voltage, and thus its unnecessary components ($C_{trap}$, $C_{dep}$, and $C_{s/d,geo}$) are normally minimized in modern MOSFET design. Note that the source/drain-to-gate fringing capacitance (including overlap capacitance) $C_{frin,s/d-g}$ is directly connected between power rails (gate to source/drain), and hence is irrelevant to central-channel potential modulation (that determines device performance) in any FETs with long-channel or with excellent electrostatic integrity. Therefore, any steep slope of fabricated NC-FETs[12] in long-channel condition should not be mis-attributed to $C_{frin,s/d-g}$. In the operation of NC-FETs, the negative $C_{NC}$ induces a negative voltage drop ($V_{NC}$) across the NC layer, as illustrated in the conduction band diagram in Fig. 1c, to compensate for the positive voltage ($V_{MOS}$) on the underlying FET, thereby potentially reducing the total gate voltage ($V_g = V_{NC} + V_{MOS}$).

**Formulating $I_d$–$V_g$ swing of a generic NC-FET.** To quantify the operation mechanism of NC-FET, the slope or swing ($S$) of the entire $I_d$–$V_g$ curve of a generic NC-FET is formulated in the form of decoupled contributions from electrostatics and transport mechanism (Fig. 2a, with detailed derivations in Supplementary Note 3). $C_{MOS}$ is the total gate capacitance of the FET under the NC layer (Fig. 1b),

$$C_{MOS} = \frac{C_{div} \cdot C_{OX}}{C_{div} + C_{OX}} \quad (3)$$

$S_{TP}$ represents the thermionic transport mechanism limited $S$. The expression of $S_{TP}$ in Fig. 2a only takes into account the lowest conduction mode for conciseness, which does not affect the conclusions drawn in this paper. All numerical simulations carried out in this work rigorously include all conduction modes.

The battle field for any steep-slope device to pursue small $SS$ is the subthreshold regime where $E_f$ is well below $E_c$. In this regime, $S_{TP}$ is reduced to 60 mV per decade at room temperature (according to the expression of $S_{TP}$ in Fig. 2a)- often referred to as "Boltzmann tyranny". Moreover, $C_Q$ is negligibly small (according to Eq. (2)), and hence $C_{div}$ is reduced to $C_{div,<Vth} = C_{trap} + C_{dep} + C_{s,geo} + C_{d,geo}$, $C_{MOS}$ is reduced to $C_{MOS,<Vth} = C_{div,<Vth} \cdot C_{OX}/(C_{div,<Vth} + C_{OX})$. $S$ is reduced to $SS$, as expressed

in Fig. 2b. $m$ is called the body factor that captures the gate efficiency of the FET under the NC layer, and $A_V$ is the voltage gain introduced by the NC layer,

$$A_V = \frac{dV_{MOS}}{dV_g} = \frac{|C_{NC}|}{|C_{NC}| - C_{MOS,<Vth}} \quad (4)$$

which is expected to be larger than 1. In other words, $m$ and $A_V$ compete with each other in determining $SS$.

**NC does not help good FETs.** The detailed physics of the equation in Fig. 2b can be "visualized" in Fig. 2c. As shown, when $|C_{NC}|$ is larger than $C_{OX}$, $m$ wins the competition, and $SS > 60$ mV per decade (shaded region on the left). When $|C_{NC}|$ is smaller than $C_{OX}$, $A_V$ dominates $SS$, and enables sub-60 $SS$. However, $|C_{NC}|$ should not be smaller than $C_{MOS,<Vth}$, otherwise $A_V$ and $SS$ become smaller than zero (Eq. (4) and Fig. 2b, and the shaded region on the right in Fig. 2c), and the total gate capacitance $C_g = |C_{NC}| \cdot C_{MOS,<Vth}/(|C_{NC}| - C_{MOS,<Vth})$ (note $C_{NC} = -|C_{NC}|$) is negative, indicating that the metastable state of the NC layer (see Supplementary Fig. 2) is not well stabilized by the FET structure underneath. In the charge-voltage or current-voltage characteristics, the metastable negative $C_g$ or negative $SS$ portion is reflected as hysteresis[7], resembling that in a ferroelectric memory device[30]. As indicated in Fig. 2b, $C_{div,<Vth}$ represents the slope of $SS/60$ line. As long as $C_{div,<Vth}$ is not too large compared to $C_{OX}$, the design space of $1/|C_{NC}|$ (normally spanned by designing the NC layer thickness, $T_{NC}$) for sub-60 $SS$, i.e., between $1/C_{MOS,<Vth}$ ($= 1/C_{OX} + 1/C_{div,<Vth}$) and $1/C_{OX}$, is reasonably spacious. Meanwhile, $C_{div,<Vth}$ should not be too small either, otherwise $|C_{NC}|$ is required to be equally small to achieve steep slope (from Fig. 2b), as illustrated by the dashed line in Fig. 2c. Considering the generally large absolute value of the dielectric constant of ferroelectric materials ($|\varepsilon_{NC}|$)[7,11,30], small $|C_{NC}|$ translates to very thick NC layer, which is impractical for state-of-the-art very-large-scale-integration (VLSI) technology. Modern MOSFETs have evolved into the ultrathin-body (UTB) era, in the form of semiconductor-on-insulator (SOI)[31], Fin-FET[32], NW-FET[33], CNT-FET[33], and 2D-FET[34] et al., accompanied with the requirement of high-quality material/interface and low channel doping (to suppress impurity scattering and performance variation), all of which are the constituents of a "good" FET for optimal gate efficiency and current drive. Unfortunately, these designs result in small $C_{s/d,geo}$, $C_{dep}$, and $C_{trap}$, resulting in very small $C_{div,<Vth}$ and hence small $C_{MOS,<Vth}$ (Eq. (3)) (also verified with numerical simulation in Supplementary Note 4). In other words, NC does not help these "good" MOSFET platforms achieve steep slope.

Although required $C_{div,<Vth}$ (as discussed above) is not feasible in state-of-the-art MOSFET platforms, it can be introduced in unconventional ways. For example, in traditional bulk MOSFET technology, $C_{div,<Vth}$ can be realized in the form of high doping induced $C_{dep}$; In SOI structure, the buried oxide (BOX) can be designed to be comparable with gate oxide in terms of capacitance value, thereby serving as an effective $C_{dep}$; In poorly designed ultra-short-channel devices and/or devices with poor interface, $C_{s/d,geo}$ and/or $C_{trap}$ can, although in undesired ways, supply an appreciable $C_{div,<Vth}$.

**Quantum capacitance may "kill" NC-FETs.** Although, as discussed above, the tuning of $C_{div,<Vth}$ is non-trivial, the greatest challenge of NC-FETs arises from the rapidly increasing channel electron density (or $C_Q$ in Eq. (2)) with gate bias near and above threshold (Supplementary Note 5), significantly increasing $C_{div}$, which is the slope of $S/S_{TP}$ versus $1/|C_{NC}|$ (see the equation in Fig. 2a). Increased $C_{div}$ is equivalent to clockwise rotating the

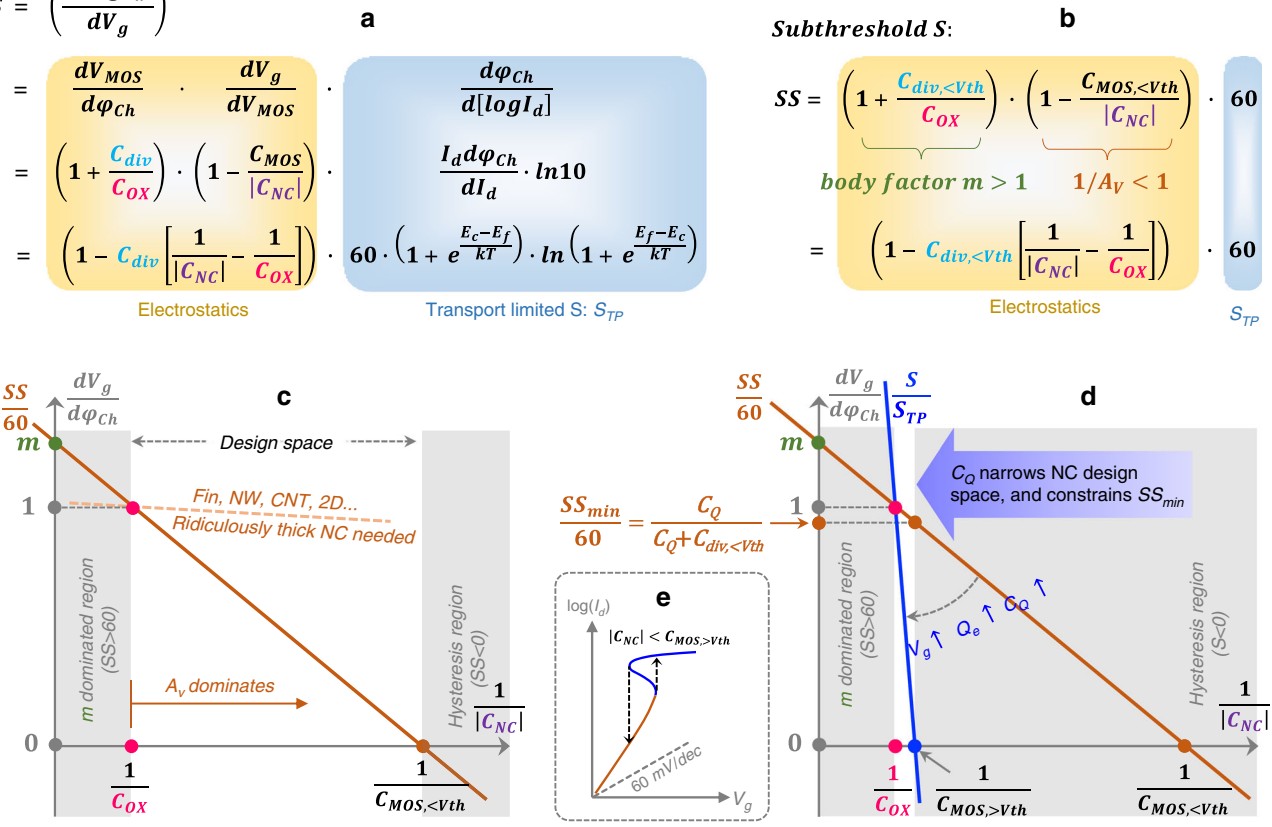

**Fig. 2 Impact of electrostatics and quantum capacitance on NC-FET design space. a** Formulating $I_d$-$V_g$ slope or swing ($S$) by decoupling the electrostatics and transport contributions. $S_{TP}$ is the transport limited $S$. **b** $S$ is reduced to subthreshold $S$ ($SS$) in the subthreshold regime. $S_{TP}$ is reduced to 60 mV per decade. Body factor $m$ (>1) and voltage gain $A_V$ (>1) compete with each other in determining $SS$. **c** Visualizing the $SS$ formula and its physics. In the shaded region on the LHS, $m$ dominates, thus $SS$ > 60 mV per decade. In the shaded region on the RHS, $SS$ < 0, represents the appearance of hysteresis. The design space here is between the two shaded regions where $A_V$ dominates and 0 < $SS$ < 60. The dashed line illustrates the ultra-small negative slope of $SS$/60 line (which implies that $SS$/60 can be noticeably below 1 only for $1/|C_{NC}| \gg 1/C_{ox}$) for Fin-, nanowire (NW)-, carbon nanotube (CNT)- and 2D- FETs, indicating very small $|C_{NC}|$, or equivalently very thick NC layer in these devices is required to obtain NC benefit. **d** A picture that captures all essential quasi-static device physics of NC-FETs. Note that in the near and above-threshold region, the $SS$/60 line transforms to the $S/S_{TP}$ line, which rotates clockwise due to the rapidly increasing quantum-capacitance $C_Q$ with gate bias till a nearly vertical position is reached when the device is fully turned on. Since $S$ in the entire range of $I_d$-$V_g$ curve should be guaranteed to be positive, in order to prevent hysteresis in both sub- and super- threshold regimes, the lower bound of $1/|C_{NC}|$ is extended to $1/C_{MOS,>Vth}$, leading to a narrow NC design space and a minimum hysteresis-free $SS$ (= $C_Q/(C_Q + C_{div,<Vth})$) at $1/|C_{NC}| = 1/C_{MOS,>Vth}$. **e** $I_d$-$V_g$ curve of an example NC-FET in which $1/|C_{NC}|$ is designed to be within the shaded region on the RHS of (**d**). Although $SS$ (brown colored portion) is small, negative $S$ (i.e., hysteresis) appears in the near- or above-threshold region (blue colored portion).

$S/S_{TP}$ line w.r.t. the $SS$/60 line, as illustrated in Fig. 2d. In order to guarantee hysteresis-free $I$-$V$ characteristics, or positive $S$, not only in the subthreshold regime, but also near- and above-threshold regimes, the lower bound of the forbidden hysteresis region (shaded region on the RHS of Fig. 2d) of $1/|C_{NC}|$ gets extended from $1/C_{MOS,<Vth}$ ($= 1/C_{div,<Vth} + 1/C_{OX}$) to $1/C_{MOS,>Vth} = (1/C_{div,>Vth} + 1/C_{OX} \approx 1/C_Q + 1/C_{OX})$, which is very close to $1/C_{OX}$, since $C_Q$ in near- and above-threshold regimes generally becomes much larger than $C_{OX}$ in typical Si or Ge MOSFETs. In other words, the design space of NC is significantly narrowed by $C_Q$. If $|C_{NC}|$ is designed to be smaller than $C_{MOS,>Vth}$, although $SS$ can be noticeably lower than 60 mV per decade, a hysteresis appears at the near- or above-threshold regime, as schematically illustrated in Fig. 2e, ending up with a memory device rather than a logic device. In the design of a FET structured FE memory[30], in which hysteresis is a desired property, the oxide layer is normally made very thin, or completely removed if the employed FE material is a good insulator. Interestingly, the reason for these devices always exhibiting hysteresis becomes very clear from Fig. 2d. With finite oxide layer thickness, if $1/|C_{NC}|$ is designed to

be within the "$m$ dominated region" (shaded region on the LHS of Fig. 2d), $S$ in the entire range of $I_d$-$V_g$ curve is guaranteed to be positive, i.e., $I_d$-$V_g$ curve is free of hysteresis. On the other hand, in the absence of an oxide layer (i.e., $1/C_{OX} \approx 0$), the "$m$ dominated region" disappears, leaving negligible $1/|C_{NC}|$ design space (between 0 and $1/C_{MOS,>Vth} \approx 1/C_Q$) in which hysteresis is absent. Hence, in the feasible $1/|C_{NC}|$ range, $S$ is always smaller than 0, which guarantees hysteresis. Figure 2d also gives the minimum $SS$ without suffering from hysteresis to be

$$SS_{min} = 60 \cdot \frac{C_Q}{C_Q + C_{div,<Vth}} \tag{5}$$

at $|C_{NC}| = C_{MOS,>Vth}$ (see derivation in Supplementary Note 6).

**Designing NC-FET in the quantum-capacitance limit.** Inspired by Eq. (5), lowering $C_Q$, desirably into the quantum-capacitance limit ($C_Q < C_{OX}$), seems to be an effective direction in enlarging the NC design space and reducing $SS_{min}$. According to the dependence of $C_Q$ on DOS (Eq. (2)), low-DOS material systems

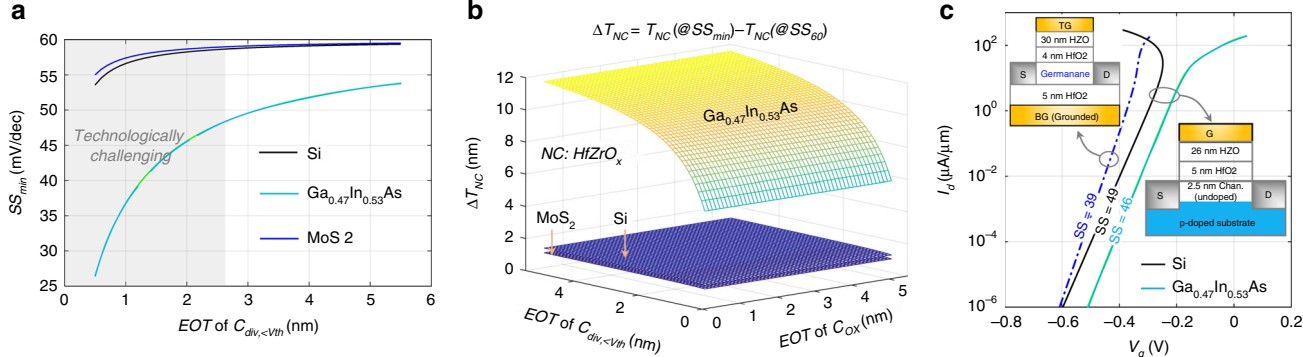

**Fig. 3 Impact of density-of-states on $SS_{min}$ and NC design space. a** $SS_{min}$ at room temperature and **b** NC thickness design space ($\Delta T_{NC}$) for Si, $Ga_{0.47}In_{0.53}As$, and monolayer $MoS_2$. The effective DOS (conduction band) of Si and $Ga_{0.47}In_{0.53}As$ are ~2.8 × 10$^{19}$ cm$^{-3}$ and ~2.1 × 10$^{17}$ cm$^{-3}$, respectively. The DOS of single-layer $MoS_2$ is ~2.4 × 10$^{14}$ eV$^{-1}$·cm$^{-2}$. The low-*DOS* $Ga_{0.47}In_{0.53}As$ exhibits significant advantages. $C_{div,<Vth}$ and $C_{OX}$ are normalized w.r.t the equivalent oxide thickness (EOT). **c** With the same device structure/size, $Ga_{0.47}In_{0.53}As$ device can (in principle) achieve smaller $SS$, w.r.t. Si device, without hysteresis. $SS$ of monolayer Germanane device can reach 39 mV per decade due to its low-DOS. The DOS of single-layer Germanane is ~2.9 × 10$^{13}$ eV$^{-1}$·cm$^{-2}$.

could help. In order to illustrate this, $SS_{min}$ is evaluated for Si (as reference), $Ga_{0.47}In_{0.53}As$, and monolayer $MoS_2$, as shown in Fig. 3a, versus $C_{div,<Vth}$, which is normalized w.r.t the equivalent oxide thickness (EOT) for the convenience of comparison. If not specified otherwise, $Hf_{0.5}Zr_{0.5}O_2$ (parameters are provided in Supplementary Fig. 2) is used as the NC material in the calculation throughout this work. As shown, the low-DOS $Ga_{0.47}In_{0.53}As$ can provide much smaller $SS_{min}$, w.r.t. Si and $MoS_2$. However, it is technologically rather difficult to implement large $C_{div,<Vth}$ with EOT less than 3 nm (i.e., close to $C_{OX}$ in state-of-the-art MOSFETs), thus hysteresis-free sub-60 $SS$ and sub-50 $SS$ can hardly be achieved in Si/$MoS_2$ and $Ga_{0.47}In_{0.53}As$-based NC-FETs, respectively.

Figure 3b shows the NC thickness design space ($\Delta T_{NC}$, essentially the width of the design space in Fig. 2d) in terms of normalized $C_{div,<Vth}$ and $C_{OX}$. $\Delta T_{NC}$ of Si and $MoS_2$ based NC-FETs almost vanish due to their large DOS, which is consistent with their near-60 $SS_{min}$ shown in Fig. 3a. The low-DOS of $Ga_{0.47}In_{0.53}As$ can help derive a feasible $\Delta T_{NC}$ of ~12 nm due to its low $C_Q$. Further reduction of the DOS of GaInAs system by increasing the In content is possible, but will result in lowered bandgap that degrades ON-OFF current ratio. To further confirm the effect of DOS and hence $C_Q$, numerical simulations are carried out to obtain the $I_d$–$V_g$ curves of Si and $Ga_{0.47}In_{0.53}As$ based NC-FETs, as shown in Fig. 3c. $C_{div,<Vth}$ in bulk Si and $Ga_{0.47}In_{0.53}As$ devices are intentionally implemented as $C_{dep}$ with an engineered doping profile as illustrated in the inset figure on the RHS. With exactly the same device structure/size, $Ga_{0.47}In_{0.53}As$ device shows a hysteresis-free $SS$ of ~46 mV per decade, while Si device shows a $SS$ of ~49 mV per decade suffering from hysteresis, which is consistent with the predicted hysteresis-free $SS_{min}$ in Fig. 3a. The thinness of 2D material is intrinsically beneficial in terms of achieving low DOS. However, the most widely studied 2D semiconductor, $MoS_2$, although is promising for short-channel MOSFETs[34], suffers due to its large electron effective mass (~0.6$m_0$) that results in high DOS. In contrast, Germanane (also a single-layer 2D semiconductor) possesses a much smaller electron effective mass (~0.07$m_0$) and a finite bandgap of ~1.5 eV[35], thus can be an ideal channel material for NC-FET design. Figure 3c also shows simulated $I_d$–$V_g$ curve of a Germanane NC-FET. The atomic scale thickness of Germanane, as well as the lack of effective doping techniques prevent Germanane based NC-FETs from exploiting $C_{dep}$. Instead, a $HfO_2$ based BOX layer is employed to act as an effective $C_{dep}$, as

illustrated in the inset on the LHS of Fig. 3c. By optimizing the oxide and NC thicknesses, $SS$ of Germanane based NC-FETs can be as low as 39 mV per decade.

**The role of NC non-linearity.** It is worthwhile to mention that the *P*–*E* characteristics in the NC region of a FE material is not strictly linear, as shown in Supplementary Fig. 2b, i.e., $|C_{NC}|$ is not a constant with bias or charge density. Using the analysis developed above, it is found that this secondary effect can slightly enlarge the NC design space, and may help a bit in deriving smaller $SS_{min}$ (Supplementary Note 7).

**IMG: Borrow parasitic charge for polarization in NC.** An internal metal gate (IMG) has been proposed to be inserted between the NC layer and the underlying MOSFET. It has been found that the IMG induced fringing capacitance (that helps induce further polarization in the NC layer) can help improve $SS$[21]. The physical mechanism of this effect is well captured by the developed analysis in this work (Supplementary Note 8). Note that although IMG is beneficial to the performance of NC-FETs from a capacitance point of view, it will introduce floating gate effect and domain formation that may destabilize NC[18].

**A practical role of NC for FETs: Voltage-loss saver.** As discussed above, although hysteresis-free sub-60 mV per decade $SS$, can be achieved in NC-FETs in principle, judicious device structural design for a proper $C_{div,<Vth}$, rigorous matching between NC and the total gate capacitance ($C_{MOS}$), which involves a strongly bias-dependent $C_Q$, and non-trivial fabrication efforts are required to demonstrate a prototype NC-FET. Moreover, there is a critical thickness for ferroelectric materials (~5 nm for $BiFeO_3$)[36], below which the ferroelectricity begins to degrade, and hence $|\varepsilon_{NC}|$ decreases, leading to reduced $|C_{NC}|$. Such a thickness dependence of ferroelectricity introduces additional difficulty to precisely match $C_{MOS,>Vth}$ in the narrow design space, as illustrated in Fig. 2d. Therefore, all fabricated "NC-FETs" in which FE layers are simply inserted in the gate of Fin-FETs, NW-FETs, Tunnel-FETs, or 2D-FETs, which are not suitable for NC-FET construction, are unlikely to be genuine NC-FETs. Moreover, almost all of the observed steep $SS$s in those fabricated "NC-FETs"[9–12] appear at low drain current level (close to the OFF current and/or current noise floor) only, which not only obscures their prospects, but also, again, casts doubt on their claims of

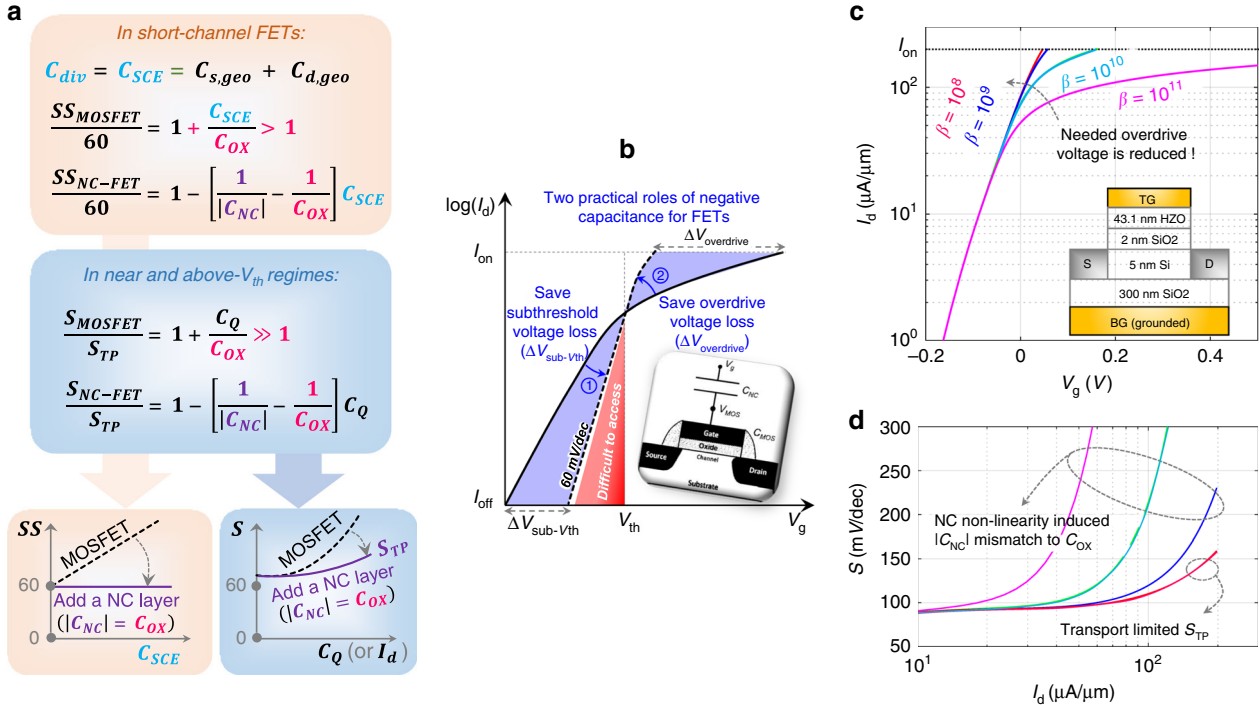

**Fig. 4 Role for NC in MOSFETs as a voltage-loss saver. a** With a simple $|C_{NC}| = C_{OX}$ matching, $SS$ can be restored to 60 mV per decade in short-channel MOSFETs, and $S$ in near- and above- threshold regimes can be reduced to the transport limit ($S_{TP}$). Here $C_{trap}$ and $C_{dep}$ are neglected, since they have been greatly suppressed in state-of-the-art MOSFETs. Thereby, **b** NC layer can help suppress the short-channel effect induced subthreshold voltage loss ($\Delta V_{sub-Vth}$), and inversion charge screening induced overdrive voltage loss ($\Delta V_{overdrive}$). It is difficult for NC to help obtain hysteresis-free steep-slope turn-on characteristics. **c** Simulated $I_d$–$V_g$ and **d** $S$ versus $I_d$ curve of a Si SOI NC-FET in which $|C_{NC}|$ is matched to $C_{OX}$. Lower NC non-linearity (quantified by the factor of $\beta$) is desired, in terms of restoring $S$ back to $S_{TP}$.

being NC-FETs, since those observations are not consistent with or supported by the fundamental physics-based predictions of the subthreshold behavior of NC-FETs in this work and other relevant work in the literature. In fact, more and more studies[37–40] have indicated that the steep $SS$s observed in many of those fabricated "NC-FETs" can be attributed to the transient effects (not captured in this work because of the ideal steady-state model employed, aimed at uncovering the essential physics and evaluating the upper-limit of NC-FET performance) during the measurement and/or ferroelectric polarization dynamics, instead of the negative capacitance effect through capacitance matching, which provide further support to the analysis and the insightful conclusions of this work, albeit from a different perspective. Here, based on the $SS$ physics of NC-FETs developed above, we also provide our insight/suggestion to those who are trying to interpret the observed steep $SS$s in various reported "NC-FETs" from transient effect point of view. Essentially, to achieve sub-thermionic $SS$, the polarization dynamics should allow FE layer in the subthreshold regime to be greatly depolarized and allow the weak residual polarization to be self-sustained (i.e., $D \approx Q_{MOS,<Vth} \approx 0$ or $P \approx -\varepsilon_0 E$), instead of relying on the charge compensation (i.e., $\varepsilon_0 E \approx 0$ or $P \approx D \approx Q_{MOS,<Vth}$) from the FET underneath, since in modern FETs, $C_{div,<Vth}$ and $C_{MOS,<Vth}$ are negligibly small. More details are provided in Supplementary Note 9. It is worth noting that such transient effects induced steep $SS$s, impose strict constraints on bias sweeping rate/direction, and operating frequency[38,40], and therefore are of limited practical use in logic transistor applications, and hence, irrelevant to the discussion in this work. It is also worthwhile to mention that the polarization response of ferroelectric materials must be enhanced[41,42] to allow the NC to function effectively at high frequencies (GHz range).

In fact, in terms of saving switching energy, NC-FET does not have to be designed as a steep-slope device. As indicated in Fig. 2d, the $S/S_{TP}$ line rotates with bias about a constant (bias and NC material independent) point ($1/C_{OX}$, 1), which provides an important implication for a new role NC can play. To make it clearer, $SS$ formula for short-channel MOSFETs and NC-FETs, and $S$ formula in the near and above-threshold regimes for MOSFETs and NC-FETs of any scale, are reorganized in Fig. 4a. In the short-channel MOSFETs, $C_{div}$ is mainly attributed to short-channel effect (SCE), specifically, $C_{div} = C_{SCE} = C_{s,geo} + C_{d,geo}$. $C_{SCE}$ prevents gate voltage in subthreshold regime to drop entirely in the channel, i.e., it causes a subthreshold gate voltage loss on the gate oxide, which is reflected by a poor $SS$, $SS_{MOSFET} = 60(1 + C_{SCE}/C_{OX}) > 60$ (see detailed derivations in Supplementary Note 3). On the other hand, in near and above-threshold regimes of MOSFETs, large $C_Q$, or electron charge screening effect, forces $V_g - V_{th}$, which is usually referred to as the overdrive voltage, to drop on the gate oxide, instead of in the channel, leading to a poor gate efficiency, or equivalently a large $S$, $S_{MOSFET} = S_{TP}(1 + C_Q/C_{OX}) \gg S_{TP}$ (see detailed derivations in Supplementary Note 3). In this sense, overdrive voltage is a type of voltage loss. Interestingly, by adding a NC layer on MOSFETs, and simply matching $|C_{NC}|$ to the constant $C_{OX}$, $C_{SCE}$, and $C_Q$ can be absorbed. In other words, $SS$ in short-channel devices can be restored to 60 mV per decade, and $S$ in the near- and above-threshold regimes can be restored to transport mechanism limited $S_{TP}$, as illustrated in the schematics in Fig. 4a, b. The significance of the latter is that it provides an alternative solution to reduce the supply voltage and hence the switching energy, i.e., by saving overdrive voltage, which also occupies a big portion of the entire supply voltage, instead of struggling to save subthreshold voltage. This idea is supported by the numerical simulation results shown in Fig. 4c, d. Note that small NC non-

linearity (proportional to the $\beta$ coefficient that is discussed in detail in Supplementary Note 7) is desired to reduce $S$ in near- and above-threshold regimes. Compared to matching $|C_{NC}|$ to the variable $C_{MOS}$ for sub-60 mV per decade $SS$, which is rather difficult to achieve and inevitably introduces variation issues, matching $|C_{NC}|$ to the constant $C_{OX}$ is obviously much more practical in terms of saving switching energy. In several reported NC-FET experiments[10,11,43,44], a common observation is that the use of the FE layer helps improve SS (instead of achieving steep slope, i.e., < 60 mV per decade), and enhance the ON-current, which very likely stems from the voltage loss saving role of NC uncovered in this paper.

## Discussion

In summary, in a very simple and generic manner, our analysis not only unambiguously clarifies the intriguing physics that limits the capability of NC-FETs in achieving steep slope, thereby invalidating all the claimed "NC-FETs" in the literature, but also helps uncover the voltage loss saving capability of NC in modern X-FETs (where X = SOI/Fin/NW/2D/1D), provided the intrinsic limitation[41,42] of ferroelectric polarization switching speed can be overcome, along with reliability issues arising from high electric-field in the DE and large interface trap density between the FE and DE. Our insightful analysis and findings provide invaluable guidance in terms of designing NC-FETs that could not only prevent further generation of scientifically misleading claims, but also prevent wasting millions of dollars of research and development expenditures and perhaps help identify a practical low-operation-voltage device that genuinely exploits NC.

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

## Acknowledgements

The research outlined in this article from the Nanoelectronics Research Lab at UCSB was supported by the ARO (grant W911NF1810366), JST CREST program (grant SB180064), and Intel Corporation.

## Author contributions

W.C. and K.B. wrote the manuscript.

## Competing interests

The authors declare no competing interests.
