## [Peer Review File · Nature Communications]

Reviewers' comments:

Reviewer #1 (Remarks to the Author):

This article provides an insightful investigation on NC mechanism. I think authors clearly point out the issues on NC switching of transistor and give a detailed discussion. I have no further comments and recommend this paper to be published in this version.

Reviewer #2 (Remarks to the Author):

The analysis of operation of NCFETs in the hysteretic steep slope, non-hysteretic steep slope and non steep slope has been studied extensively and published before.

D J Frank et al, The Quantum Metal Ferroelectric Field-Effect Transistor, IEEE TRANSACTIONS ON ELECTRON DEVICES, VOL. 61, NO. 6, JUNE 2014 2145

Masaharu Kobayashi and Toshiro Hiramoto, On device design for steep-slope negative-capacitance field-effect-transistor operating at sub-0.2V supply voltage with ferroelectric HfO₂ thin film, AIP Advances 6, 025113 (2016);

It is well-known that for capacitance matching below inversion, low density of states material and with low NC capacitance (with impractically thick FE), hysteretic free NCFETs can be designed but with low current. The idea of using Germanane to demonstrate a steep slope NCFET below V_t makes sense. However, germanane is thermally unstable material and hence impractical.

It is also well-known that for capacitance matching above threshold, having a high density of states (DOS) material and having C_{ox} match the CNC makes sense. But this is again quite well-known as was pointed out by DJ Frank et al.

The authors point out correctly the design space of NCFETs, but the work is incremental and not novel enough to warrant publication in this journal.

Reviewer #3 (Remarks to the Author):

The authors discuss a formulation that endeavors to explain various modes of operation of MOSFET devices with ferroelectric gates. The key result from the study sheds light on how ferroelectric gate oxide thickness can be tuned to obtain various features (steep SS, hysteresis, etc) in gate control of the channel. The work is insightful, clear and crisp; however it may be important to describe some nuances associated with ferroelectric gated structures in better detail and for completeness.

1. The abstract is lacking important results/learnings obtained in the paper. Please add simple abstracts of findings on 'well-designed' scaled MOSFETs into the abstract as that is a very valuable highlight of the paper.

2. Authors mention that thinning of ferroelectric may increase C_{NC} , however, once a certain critical thickness is reached, the C_{NC} decreases. While the model only treats the device at the capacitance level (without going into what realistic thicknesses can result in that capacitance), it may be very insightful to explain how experimental effects translate to various points on the optimization curve in Fig. 2.

3. Regarding the non-linearity referenced in the paper on Page 9 (and S7), the authors present a

clear analysis of damping effect. At short channel lengths, the non-linearity can arise from discrete switching of few domains which may have multiple individual voltages, each with slightly different P-E slope or non-linearity term (which aggregates to the final average discussed here). As we know, today's scaled transistors get severely loaded by 'bias-dependent' C_{gs} and C_{gd} and C_{miller} . Will these capacitances be affected by discrete nature of ferroelectric polarization? Can the authors comment on this aspect in the supplementary information?

4. In the first statement on Page 10, do the authors imply that transient effects and associated NC benefits are somehow limited in operating frequency? This seems to be an assumption without any literature/supporting evidence at the correct scaled dimensions and loading conditions. Please clarify if the steady-state model of attached evidence has any bearing on this.

Overall, the authors present a yeoman analysis of the device using clear and lucid formulations that will be extremely valuable in designing MOSFETs with ferroelectric gates.

Response to the Reviewers' Comments on Manuscript NCOMMS-19-18805

We thank all three reviewers for their time and efforts in reviewing and commenting on our manuscript. Here we provide a point-by-point response to their comments. Corresponding revisions in the main manuscript and the supplementary information file have been highlighted in **BLUE** for the convenience of the reviewers.

Reviewers' comments:

Reviewer #1 (Remarks to the Author):

This article provides an insightful investigation on NC mechanism. I think authors clearly point out the issues on NC switching of transistor and give a detailed discussion. I have no further comments and recommend this paper to be published in this version.

Response: We thank the reviewer for his/her appreciation of our work and recommendation for publication.

Reviewer #2 (Remarks to the Author):

The analysis of operation of NCFETs in the hysteretic steep slope, non-hysteretic steep slope and non steep slope has been studied extensively and published before.

Response: Firstly, we appreciate that the reviewer acknowledged the correctness of our work in his/her last comment. We also thank the reviewer for citing the two papers, albeit to illustrate existing “well-known” knowledge. However, we beg to differ. We are well aware of these two articles along with several others that discuss hysteresis/non-hysteresis behavior of NCFETs using device simulations, which however, **offer little physical insight and thereby have not been able to help the relevant device designers**. This is clearly evident from the fact that during the past three-to-five years since the publication dates (one in 2014 and the other in 2016) of the two papers listed by the reviewer, numerous (>>100) “NCFETs” have been designed/fabricated in those incorrect ways as illustrated in our manuscript. Interestingly, many of them were published in high-impact and/or professional electron device journals/conferences such as *Nature Nanotechnology* (Ref. 8 in the original main manuscript), *Nature Communications* (see Ref. [1] below), *Nano Letters* (see Refs. [2][3] below), *IEEE Electron Device Letters* (Refs. 5 and 7 in the original main manuscript), *IEEE International Electron Device Meeting* (Refs. 4 and 6 in the original main manuscript), etc. To further illustrate this point, we have now cited a few more (Refs. [1]-[3] listed below) such “NCFETs” [Refs. 13-15 in the revised main manuscript]. This fact is obviously conflicting with the statement of the reviewer that the analysis in our manuscript is well known, and is also the main reason we decided to submit our work to a high-impact journal like *Nature Communications*, in order to clarify the important design rule and limitation of NCFETs, and thereby steer the research community in the right direction. Here we provide a point-by-point response to the reviewer’s comments.

[1] X. Wang et al., “Van der Waals negative capacitance transistors,” *Nature Comm.*, vol. 10, no. 3037, 2019.

[2] M. Si et al., “Steep-slope WSe₂ negative capacitance field-effect transistor,” *Nano Lett.*, vol. 18, p. 3682, 2018.

[3] F. McGuire et al., “Sustained sub-60 mV/decade switching via the negative capacitance effect in MoS₂ transistors,” *Nano Lett.*, vol. 17, p. 4801, 2017.

D J Frank et al, The Quantum Metal Ferroelectric Field-Effect Transistor, IEEE TRANSACTIONS ON

ELECTRON DEVICES, VOL. 61, NO. 6, JUNE 2014 2145.

Response: We are very familiar with this paper. As we have discussed in the supplementary information (section S8) of our manuscript, the employment of an internal metal gate can help borrow parasitic charge from the source/drain overlap/fringing capacitance for the polarization in the ferroelectric layer, and hence realize voltage amplification. However, the electrically floating nature of the internal metal gate may lead to undesirable voltage shifting and/or additional current-voltage hysteresis in addition to that induced by NC instability, depending on the leakage currents and timing of the logic signals. **D. J. Frank and P. M. Solomon et al., in the above cited paper, proposed a modified version of the internal metal gate** to eliminate the effect of floating voltage by employing an ultrathin quantum metal. This unknown quantum metal bears stringent requirements, including rigorous control over density of states to achieve a balance between providing compensation charge for FE polarization and screening the channel potential (and hence current) modulation, as well as high dielectric constant (> 40). Unfortunately, this quantum metal has not yet been realized so far. In summary, this is an interesting paper, but is not directly relevant to the main focus of our work.

Masaharu Kobayashi and Toshiro Hiramoto, On device design for steep-slope negative-capacitance field-effect-transistor operating at sub-0.2V supply voltage with ferroelectric HfO₂ thin film, AIP Advances 6, 025113 (2016);

Response: We have been also aware of this paper that evaluates the hysteretic and non-hysteretic steep slope of a specific FET structure using **numerical simulations**, as have also been shown by many other previously published numerical simulation based NCFET papers (several of which are already cited in our manuscript such as Refs. 21, 22, 25, 28, etc), but never explained those phenomena in a lucid and uncomplicated way as we did in our work. We are sure that the reviewer is aware of the well known fact that simulations do not provide much physical insight into "why" certain phenomenon occur and the relative impact of different parameters affecting the phenomenon. That is why we need simple closed-form analytical treatment that is easy to understand and provide clear insights into the various phenomena, thereby uncovering the essential physics, design rule, and limitations, as we did in our manuscript, which has been recognized and understood by reviewer 1 and 3. Nevertheless, we have now included this paper as Ref. [24] in the revised main manuscript.

It is well-known that for capacitance matching below inversion, low density of states material and with low NC capacitance (with impractically thick FE), hysteretic free NCFETs can be designed but with low current. The idea of using Germanane to demonstrate a steep slope NCFET below V_t makes sense. However, germanane is thermally unstable material and hence impractical.

Response: We are a bit perplexed by these comments. On one hand, we find it rather strange that the reviewer states "for capacitance matching below inversion, low density of states material and with low NC capacitance (with impractically thick FE), hysteretic free NCFETs can be designed but with low current", since this is reflected in neither of the two papers he/she referred to. On the other hand, the reviewer might have missed that we clearly pointed out that one **must have a proper voltage divider capacitor**, in order to have any low density-of-states (DOS) material beneficial for broadening the design space of the minimum attainable subthreshold swing of NCFETs, instead of being simply useful for "capacitance matching below inversion, with low NC capacitance". Moreover, a simple statement that low DOS material channel will lead to low current may not be true. There is a trade-off between the number of current conduction channels (proportional to DOS) and carrier velocity (inversely proportional to the effective mass and hence to the DOS), in reaching the maximum drive current.

We thank the reviewer for appreciating our idea of using Germanane to construct NCFETs. We agree with the reviewer that Germanane is thermally unstable. However, there has been effective ways to prevent the degradation of unstable 2D materials, such as using the thermally stable and electrically insulating hexagonal boron nitride passivation layers at both surfaces. In any case, the main purpose of proposing Germanane is to simply **highlight how our analytical framework can be successfully employed to perform predictive analysis** to aid the design of genuine NCFETs.

It is also well-known that for capacitance matching above threshold, having a high density of states (DOS) material and having C_{ox} match the C_{NC} makes sense. But this is again quite well-known as was pointed out by DJ Frank et al.

Response: Again, we find the reviewer's comments rather baffling. First off, the reviewer's statement is not reflected in either our manuscript or D.J. Frank's paper. In fact, there are no clear advantages of employing high DOS materials for capacitance matching above threshold (see Equation 5 in our main manuscript). Secondly, the benefits of matching C_{ox} with C_{NC} has not been stated in any previous publications. As we pointed out in our manuscript for the very first time, the purpose of such matching is to recycle the voltage loss induced by large quantum capacitance and short-channel effects. We have carefully double-checked D.J. Frank's paper. As stated earlier, **D.J. Frank's paper discusses how to design a quantum metal to replace the internal metal gate in NCFETs**, in terms of reaching a balance between providing necessary compensation charge for the polarization in ferroelectric layer and screening the channel potential (and current) modulation, which has nothing to do with the density of states of the channel that we discussed in our manuscript. Also, "having C_{ox} match the C_{NC} " to recycle the subthreshold voltage loss in short-channel devices and the above-threshold overdrive voltage loss in any FETs is not mentioned anywhere in D.J. Frank's paper.

The authors point out correctly the design space of NCFETs, but the work is incremental and not novel enough to warrant publication in this journal.

Response: We thank the reviewer for acknowledging the correctness of our work. As already pointed out in our response to the reviewer's comments above, the unique and clear analytical treatment formulated in our work stands to benefit the entire device community in terms of knowledge required to design a genuine NCFET, as well as practical utilization of NC in CMOS transistors, which represents a significant contribution given the imminent need to scale the supply voltage for lowering power consumption of CMOS as well as to steer the rapidly growing NCFET community in the right direction. We'd like to re-emphasize that **our analytical formulation of the device physics of NCFETs, and three main conclusions:** "1) the excellent electrostatic integrity of state-of-the-art MOSFET platforms, such as SOI-FET, FinFET, nanowire-FET, carbon nanotube (CNT)-FET, and two-dimensional (2D) material based FETs, prevents their benefiting from NC in terms of achieving small SS or low voltage operation; 2) the quantum capacitance is the limiting factor for NC-FETs to achieve hysteresis-free subthermionic SS, and FETs that can operate in the quantum capacitance limit are desired platforms for NC-FET construction; and 3) compared to the target of achieving subthermionic SS, a more practical role of NC in FETs is to recycle the subthreshold voltage loss caused by short-channel effects, and the overdrive voltage loss caused by the large quantum capacitance;" **have never been reported** in any previous papers.

To summarize, as already pointed out to the reviewer, **our physical yet lucid analysis** provides the device designers a unique predictive ability and device optimization framework that should be extremely valuable in designing MOSFETs with ferroelectric gates, as recognized by the other two reviewers.

Reviewer #3 (Remarks to the Author):

The authors discuss a formulation that endeavors to explain various modes of operation of MOSFET devices with ferroelectric gates. The key result from the study sheds light on how ferroelectric gate oxide thickness can be tuned to obtain various features (steep SS, hysteresis, etc) in gate control of the channel. The work is insightful, clear and crisp; however it may be important to describe some nuances associated with ferroelectric gated structures in better detail and for completeness.

Response: We thank the reviewer for his/her careful reading of our manuscript and for appreciating the significance and novelty of our contribution, as well as for the constructive suggestions.

1. The abstract is lacking important results/learnings obtained in the paper. Please add simple abstracts of findings on ‘well-designed’ scaled MOSFETs into the abstract as that is a very valuable highlight of the paper.

Response: We thank the reviewer for his/her constructive suggestion. If we understand the reviewer’s suggestion correctly, the reviewer is referring to the requirements to be a “well-designed” scaled MOSFET. We are definitely willing to add it into the abstract. Unfortunately, after we received the comments from reviewers, we also received a notice from the Nature Comm. editorial office that the abstract length of Nature Comm. manuscripts should be no more than 150 words. Therefore, we have to shrink our original abstract significantly. However, we have still added a simple description in the revised abstract on well-designed MOSFETs. On the other hand, in the revised main manuscript, the last paragraph of the Introduction section also summarizes our findings on well-designed scaled MOSFETs.

2. Authors mention that thinning of ferroelectric may increase C_{NC} , however, once a certain critical thickness is reached, the C_{NC} decreases. While the model only treats the device at the capacitance level (without going into what realistic thicknesses can result in that capacitance), it may be very insightful to explain how experimental effects translate to various points on the optimization curve in Fig. 2.

Response: We thank the reviewer for his/her suggestion. If we understand the reviewer’s suggestion correctly, the reviewer is referring to the thickness limit of ferroelectric material to preserve the ferroelectricity (Ref. [4] below). As the thickness of ferroelectric layer decreases, the absolute value ($|C_{NC}|$) normally increases following the capacitance formula, $|C_{NC}| = |\epsilon_{NC}|/T_{NC}$. However, when the thickness decrease below a certain critical value (~5 nm for BiFeO₃), the ferroelectricity begins to degrade, and hence $|\epsilon_{NC}|$ decreases, leading to reduced $|C_{NC}|$. In the revised manuscript, we have added relevant discussion of this phenomenon and made a correspondence to Figure 2.

[4] J. Steffes et al., “Thickness scaling of ferroelectricity in BiFeO₃ by tomographic atomic force microscopy,” *PNAS*, vol. 116, pp. 2413-2418, 2019.

3. Regarding the non-linearity referenced in the paper on Page 9 (and S7), the authors present a clear analysis of damping effect. At short channel lengths, the non-linearity can arise from discrete switching of few domains which may have multiple individual voltages, each with slightly different P-E slope or non-linearity term (which aggregates to the final average discussed here). As we know, today’s scaled transistors get severely loaded by ‘bias-dependent’ C_{gs} and C_{gd} and C_{miller} . Will these capacitances be

affected by discrete nature of ferroelectric polarization? Can the authors comment on this aspect in the supplementary information?

Response: We totally agree with the reviewer that the multi-domain formation in general ferroelectric materials also lead to non-linearity, and will impact the device behavior in scaled devices. In the supplementary information of the revised manuscript, we have added relevant comments to clarify this issue.

4. In the first statement on Page 10, do the authors imply that transient effects and associated NC benefits are somehow limited in operating frequency? This seems to be an assumption without any literature/supporting evidence at the correct scaled dimensions and loading conditions. Please clarify if the steady-state model of attached evidence has any bearing on this.

Response: Thanks for raising this issue. Yes, in fact, recent experiments (see Ref. 38 in the revised main manuscript) have uncovered that transient effects and associated NC benefits are significantly influenced by operating frequency. In the revised manuscript, we have now referred to this paper at the right place.

In this work, we aim at uncovering the essential physics and evaluating the upper-limit performance of NCFETs, and therefore consider the steady-state only. In other words, we assume that the polarization response in the ferroelectric layer is fast enough to support the device/circuit operation at any frequency, and the device test structure does not introduce any parasitic capacitance and resistance, which are one of the many sources of transient effects. Although our steady-state model does not capture those transient effects, it proves that those naively fabricated “NCFETs” are unlikely to be genuine NCFETs, which indirectly supports many recent reports (Refs. 37-40 in the revised main text) that attribute the measured steep SSs to the transient effects during the measurement and/or ferroelectric domain dynamics, instead of the negative capacitance effect through capacitance matching. Such bias sweeping rate/direction dependent steep SSs are problematic in logic application. On the other hand, both theory (see Ref. 41 in the revised main text) and experiments (see Ref. 42 in the revised main text) have found that the non-ideal (not sufficiently fast) ferroelectric response limits the operation frequency to few MHz within which the NC can help achieve voltage amplification. In the revised manuscript, we have clarified this point and added relevant references.

Overall, the authors present a yeoman analysis of the device using clear and lucid formulations that will be extremely valuable in designing MOSFETs with ferroelectric gates.

Response: Again, we thank the reviewer for his/her meticulous examination of our work. The reviewer’s comments were indeed helpful in improving the clarity of our work.

REVIEWERS' COMMENTS:

Reviewer #3 (Remarks to the Author):

The reviewer is satisfied with the responses given by the authors. The reviewer does not think that this paper alone answers the fundamental question about negative capacitance and its performance gains in a logic technology. However, it does give a clear documentation of an analytical based approach on NC device design. The reviewer hopes that authors could have a more detailed transient model with data-driven phenomena.